# Exploring the Link between *Helicobacter pylori*, Gastric Microbiota and Gastric Cancer

**DOI:** 10.3390/antibiotics13060484

**Published:** 2024-05-24

**Authors:** Costantino Sgamato, Alba Rocco, Debora Compare, Kateryna Priadko, Marco Romano, Gerardo Nardone

**Affiliations:** 1Gastroenterology Unit, Department of Clinical Medicine and Surgery, University Federico II of Naples, 80131 Naples, Italy; costa.sgamato@gmail.com (C.S.); debora.compare@unina.it (D.C.); nardone@unina.it (G.N.); 2Hepatogastroenterology Unit, Department of Precision Medicine, University of Campania “L. Vanvitelli”, 80138 Naples, Italy; kateryna.priadko@unicampania.it (K.P.); marco.romano@unicampania.it (M.R.)

**Keywords:** gastric cancer, gastric microbiota, *H. pylori*

## Abstract

Gastric cancer (GC) still represents one of the leading causes of cancer-related mortality and is a major public health issue worldwide. Understanding the etiopathogenetic mechanisms behind GC development holds immense potential to revolutionize patients’ treatment and prognosis. Within the complex web of genetic predispositions and environmental factors, the connection between *Helicobacter pylori* (*H. pylori*) and gastric microbiota emerges as a focus of intense research investigation. According to the most recent hypotheses, *H. pylori* triggers inflammatory responses and molecular alterations in gastric mucosa, while non-*Helicobacter* microbiota modulates disease progression. In this review, we analyze the current state of the literature on the relationship between *H. pylori* and non-*Helicobacter* gastric microbiota in gastric carcinogenesis, highlighting the mechanisms by which microecological dysbiosis can contribute to the malignant transformation of the mucosa.

## 1. Introduction

Gastric cancer (GC) ranks as the fifth most common cancer and the third leading cause of cancer-related death [1]. *Helicobacter pylori* (*H. pylori*) is the most well-established risk factor for GC, being associated with an up to six-fold increased risk of non-cardia adenocarcinoma [2,3]. However, only 1–3% of the infected individuals progress to GC [4,5,6]. The genetic characteristics of both the bacterium [7,8,9,10] and the host [11,12,13], as well as environmental factors, such as diet and lifestyle, can impact *H. pylori*-associated disease outcomes [14,15]. However, none of these factors fully account for GC risk, and it is not yet possible to predict who will develop cancer.

Increasing evidence suggests a potential role of non-*Helicobacter* gastric microbiota as a significant contributor to GC development [16]. This hypothesis stems from the observation that, at a certain point, gastric carcinogenesis appears to occur independently of the presence of *H. pylori*. Indeed, *H. pylori* colonization decreases over time in individuals with precancerous gastric lesions and eventually disappears in the adenocarcinoma stage [17]. The hypo/achlorhydria of the gastric environment induced by the chronic *H. pylori* infection shifts the gastric microbiota composition toward microbes typically found in the oral cavity and lower bowel, which would not normally thrive in acidic environments. The altered microbiota may promote the malignant transformation of gastric mucosa by sustaining inflammation, increasing the production of carcinogenic N-nitroso compounds and numerous other pro-carcinogenic mechanisms [18,19]. 

From this point of view, *H. pylori* might function through a hit-and-run mechanism, predisposing the gastric mucosa to subsequent oncogenic alterations, which other microorganisms could facilitate [20]. 

In this review, we synthesize the most recent evidence on the microecological dysbiosis in gastric carcinogenesis, highlighting the mechanisms by which *H. pylori* and the non-*Helicobacter* gastric microbiota cooperate in the malignant transformation of the mucosa. 

## 2. The Gastric Microbiota in Healthy Individuals

The discovery of *Campylobacter pyloridis* in 1982 demonstrated that bacteria could survive and thrive in the stomach’s acidic environment, definitively shifting the paradigm of the stomach as a sterile niche for bacteria [21]. Later, advancements in high-throughput and next-generation sequencing techniques confirmed that the stomach harbors a complex microbial community typical of the organ and different from that found in other parts of the digestive tract.

The gastric microbiota of healthy individuals comprises 57 bacterial genera across eight phyla, including *Proteobacteria*, *Firmicutes*, *Bacteroidetes*, *Actinobacteria*, *Fusobacteria*, *Spirochetes*, *Tenericutes*, and *Saccharibacteria*. The most prevalent genera for each phylum are *Streptococcus* and *Veillonella* (*Firmicutes*), *Neisseria* and *Haemophilus* (*Proteobacteria*), *Fusobacterium* (*Fusobacteria*), *Prevotella* and *Porphyromonas* (*Bacteroidetes*), and *Rothia* (*Actinobacteria*) [22,23,24]. Interestingly, more than 65% of the bacterial phylotypes in the stomach also inhabit the human mouth [25,26]. 

The richness and diversity of gastric microbiota significantly vary between mucosa and gastric juice and between antrum and corpus. In particular, the microbiota inhabiting gastric juice, primarily composed of *Firmicutes*, *Bacteroidetes*, and *Actinobacteria* phyla, does not accurately represent the diversity of stomach bacteria, as transient bacteria fail to colonize the gastric mucosa [27]. Furthermore, species richness in the corpus is notably higher than in the antral mucosa, albeit with no significant difference [28].

Several factors can influence the composition of gastric microbiota [29], such as dietary patterns [30], geographical location [31], prolonged use of proton pump inhibitors [32] or probiotics [33], surgical interventions [34], and, above all, *H. pylori* infection [35]. 

In Mongolian gerbils, *H. pylori* infection was associated with the occurrence of *Enterococcus* spp. and *Staphylococcus aureus* in both the stomach and duodenum. Additionally, infected animals exhibited a notable increase in *Bacteroides* spp. in the stomach, while *Lactobacillus* spp. had a decreasing trend. Moreover, the levels and distributions of *Bifidobacterium* spp., *Bacteroides* spp., and total aerobes changed. Intriguingly, gerbils positive for *H. pylori* colonization had a higher histopathologic score for gastritis and a similar score for duodenitis [36].

In the healthy human gastric mucosa, *Streptococcus* and *Prevotella* stand out as the dominant genera [22,23,24]. However, upon infection, *H. pylori* becomes the predominant gastric microbe, prevailing in up to 40–99% of the total microbial community. This explains up to 28% of individuals’ overall variation in gastric microbiota [37,38].

*H. pylori* infection negatively correlates with the alpha diversity of gastric microbiota and effectively clusters groups of infected and uninfected individuals based on the beta diversity [39]. In particular, *H. pylori* is accompanied by a lower abundance of *Actinobacteria*, *Bacteroidetes*, and *Firmicutes* and an increase in *Proteobacteria* [40,41,42]. Furthermore, there exists a negative association between *H. pylori* and some gastric members, such as *Acidovorax*, *Aeromonas*, *Bacillus*, *Bradyrhizobium*, *Halomonas*, *Cloacibacterium*, *Meiothermus*, *Methylobacterium*, and *Ralstonia*, while the interactions of the non-*H. pylori* members are positively correlated [43]. 

## 3. *H. pylori* and Cancer

*H. pylori* infection significantly increases the risk of non-cardia adenocarcinomas with an overall odds ratio (OR) of 21.0 [44,45,46]. Persistent colonization of the stomach by *H. pylori* induces a stepwise process that can eventually progress into GC. According to the gastric precancerous cascade, *H. pylori* primarily triggers the transition from normal mucosa to non-atrophic gastritis (NAG). Subsequently, it induces a series of progressive changes, starting with the loss of stomach-specific cell types (atrophic gastritis, AG), then advancing to intestinal metaplasia–dysplasia, and finally culminating in gastric adenocarcinoma. [4,5]. Several factors may contribute to *H. pylori*-induced gastric carcinogenesis, such as strain variation, genetic polymorphisms of hosts, and environmental influences. 

Different allotypes of bacterial virulence factors, especially cytotoxin-associated gene A (CagA) and particular alleles of oligomeric vacuolating cytotoxin A (VacA), are closely associated with GC development. The CagA gene characterizes more virulent strains of *H. pylori*, increasing the risk of developing GC by 1.64-fold [47]. It encodes the CagA protein, which promotes tumorigenesis by activating inflammatory signaling and oncogenic pathways, stimulating cell proliferation, losing contact inhibition, and suppressing apoptosis [7,8,9,47]. VacA is a pore-forming cytotoxin secreted by *H pylori*, causing cell vacuolization and other cytotoxic features, such as alterations in plasma membrane permeability [10], paracellular leakage in epithelial monolayers [48], and mitochondria dysfunction [49]. VacA protein destroys the tight junctions of epithelial cells, induces apoptosis by interfering with mitochondrial function [4], and causes alteration in cell signaling [50,51]. The ability of VacA to downregulate autophagy and lysosomal degradation contributes to the accumulation of CagA in gastric epithelial cells [52]. Polymorphisms and epigenetic alterations in genes linked to both the adaptive and innate immune systems, such as interleukins (IL1β and IL8), transcription factors (CDX2, RUNX3, and TLR1), and DNA repair enzymes, can influence the individual’s susceptibility to GC development [11,12,13]. 

A Western dietary pattern, characterized by high red meat and salt intake and a low intake of antioxidant foods (fruit and vegetables), is associated with increased GC risk [14,15]. Habitual high salt intake increases susceptibility to *H. pylori* infection and can synergistically promote GC through glandular atrophy, DNA damage, and cell proliferation. Similarly, a poor-quality diet with consumption of salt-preserved smoked foods, rich in carcinogenic N-nitroso compounds, promotes cancer development [52]. A high-fat diet may induce gastric dysbiosis (characterized by an increase in *Lactobacillus* abundance), intestinal metaplasia; overexpression of leptin, phosphorylated leptin receptor, and STAT3; and intracellular b-catenin accumulation in GC [27]. Finally, other established factors associated with an increased GC risk include older age, low socioeconomic status, obesity, industrial and chemical pollutants, cigarette smoking, and heavy alcohol drinking [2].

## 4. Non-*H. pylori* Gastric Microbiota and Cancer

Robust evidence supporting the potential role of non-*H. pylori* microbiota in gastric carcinogenesis emerged from studies on animal models. 

In C57BL/6N mice infected with the same strain of *H. pylori* but raised in different environments (i.e., Charles River Lab and Taconic farms), the incidence of severe inflammation, gastritis, and metaplasia was higher in those from Charles River Lab than in those from Taconic farm. The colonization rate by *Lactobacillus* strains differed significantly between the two groups, thus suggesting that mice grown in different environments harbor distinct stomach microbiota [53]. 

In the insulin–gastrin (INS-GAS) transgenic mouse model, mice harboring a gastric-specific pathogen-free microbiota developed more severe gastric precancerous lesions than those with *H. pylori* mono-infection [54]. The colonization with altered Schaedler’s Flora (a microbiota restricted to three species, including ASF356 *Clostridium* species, ASF361 *Lactobacillus murinus*, and ASF519 *Bacteroides* spp.) was sufficient to promote gland atrophy or dysplasia in mice. However, *H. pylori* coinfection led to more severe abnormalities, and 69% of mice with gastric dysplasia had intraepithelial neoplasia. In particular, the overgrowth of *Lactobacillus murinus* ASF361 correlated with a robust expression of gastric inflammation and cancer molecules, including TNF-α, Ptger4, and TGF-β [55]. On the other hand, antibiotic treatment significantly prevented the progression of gastric neoplasia in *H. pylori*-free and specific pathogen-free INS-GAS mice [56].

The transgenic K19-Wnt1/C2mE mice (GAN mice) express Wnt1, COX-2, and PGE2 in the gastric mucosa and develop gastric tumors with histological characteristics closely resembling those observed in human intestinal-type GC. Oshima et al. observed that GAN mice raised under germ-free conditions develop significantly smaller tumors than specific pathogen-free (SPF)-GAN mice. Likewise, antibiotic treatment of SPF-GAN mice suppressed gastric tumor development. In addition, the reconstitution of commensal bacteria in germ-free (GF)-GAN mice and the infection with *Helicobacter felis* led to the development of gastric tumors [57,58]. Finally, gastric microbiota tissue and fluids transplanted from human patients with intestinal metaplasia or GC induce precancerous features in GF- *salivarius* or *Staphylococcus epidermidis* causes increased gastric damage or reduced proinflammatory cytokine responses, respectively, compared to mice infected solely with *H. pylori* [59,60]. 

Based on this evidence, it seems plausible that non-*H. pylori* bacteria could contribute to developing gastric diseases through synergistic or competitive interactions with *H. pylori* or independent of it. However, the nature of the interaction between *H. pylori* and non-*Helicobacter* microbiota in gastric carcinogenesis still remains unclear. 

One hypothesis proposes that chronic *H. pylori* infection, leading to gastric atrophy and achlorhydria, favors the overgrowth of non-*Helicobacter* species. These bacteria potentially fuel malignant transformation by perpetuating inflammation and generating carcinogenic N-nitroso compounds, including N-nitrosamines and nitric oxide, which induce DNA damage and reactive oxygen species (ROS) [61,62]. For example, *Veillonella parvula* and *Haemophilus parainfluenzae* could enhance nitrite accumulation in gastric juice due to their faster nitrate-reducing activity, causing DNA damage and methylation in epithelial cells. Similarly, *Escherichia coli*, *Streptococcus*, *Clostridium*, *Haemophilus*, *Veillonella*, *Staphylococcus*, *Neisseria*, *Nitrospirae*, and *Lactobacillus* could contribute to GC through the production of carcinogenic N-nitroso compounds [63]. Finally, lactic acid bacteria produce ROS, which induce DNA damage, reduce nitrate to nitrite, drive the oncogenes’ activation, enhance angiogenesis, and inhibit apoptosis [16].

Although data from animal studies provide biological plausibility for the hypothesis that the gastric microbiota can impact the type and timing of gastric lesions, human studies have not yielded conclusive results. The profile of the gastric microbiota at the time of diagnosis cannot provide insight into the causal role of bacteria in human GC. On the other hand, due to the long natural history of gastric diseases, conducting longitudinal studies to correlate specific bacterial signatures with various disease outcomes is not feasible. 

In Colombia, where GC reaches one of the highest incidence and prevalence rates globally, infection by *H. pylori* is nearly universal [64]. However, among inhabitants of Tumaco on the Pacific Coast, GC’s incidence and mortality rates are significantly lower than those observed in the Túquerres Mountain region. The residents of mountainous areas and coastal regions exhibit significant differences in terms of population ancestry, lifestyle, diet, and prevalence of intestinal parasite infestations. Nevertheless, none of these factors, together with the phylogeographical group of *H. pylori* or the presence of cag pathogenicity island, provide a complete rationale for the so-called “Colombian Enigma” [65]. Interestingly, several operational taxonomic units (OTUs) were detected exclusively in either the Tumaco or Tùquerres population. Specifically, some taxa, such as *Leptotrichia wadei* and *Veillonella* spp. were more abundant, and *Staphylococcus* spp. were less abundant in gastric biopsies of individuals from Túquerres than those from Tumaco, highlighting the potential role of non-*Helicobacter* gastric microbiota in the carcinogenic process [31].

When GC develops, the composition of the resident microbiota significantly differs between cancerous and non-cancerous individuals.

In one of the first studies employing terminal restriction fragment length polymorphisms alongside 16S rRNA gene cloning and sequencing, the characterization of gastric microbiota in 10 GC patients and 5 dyspeptic controls revealed a relatively low abundance of *H. pylori*. Conversely, various species from the genera *Streptococcus*, *Lactobacillus*, *Veillonella*, and *Prevotella* were predominant in the GC microbiota [66]. Castaño-Rodríguez et al. found a marked increase in the relative abundances of *Lactococcus*, *Fusobacterium*, *Veillonella*, *Haemophilus*, and *Leptochichia* in GC patients compared to control samples. Of interest, predicted pathways related to bacterial carbohydrate metabolism, such as short-chain fatty acid production, were enriched in GC [67]. A more extensive study including 103 patients with GC compared to 212 with chronic gastritis (CG) found a markedly increased bacterial load in GC, albeit with no significant diversity at the phylum level. However, five genera with potential cancer-promoting activities, namely *Lactobacillus*, *Escherichia–Shigella*, *Nitrospirae*, *Burkholderia fungorum*, and *Lachnospiraceae* uncultured, were found to be enriched in GC samples. Interestingly, *Nitrospirae* was found in all GC patients and was absent in CG cases [68].

The microbial profiling of the gastric mucosa of 54 patients with GC compared to 81 patients with CG revealed reduced bacterial diversity and *Helicobacter* abundance in GC cases. Conversely, commensal genera such as *Achromobacter*, *Citrobacter*, *Clostridium*, *Lactobacillus*, and *Rhodococcus* were over-represented. The combination of ten relevant bacteria with a different abundance in GC and CG established a microbial dysbiosis index that demonstrated a strong performance in effectively distinguishing CG and GC cases. Interestingly, the microbial communities identified in GC had nitrosation functions, indicative of increased genotoxic potential [17]. Coker et al. obtained similar results by identifying five non-*H. pylori* bacterial taxa of oral origin (*Peptostreptococcus stomatis*, *Streptococcus anginosus*, *Parvimonas micra*, *Slackia exigua*, and *Dialister pneumosintes*), which exhibited significant centrality in the GC microbial ecological network and distinguished GC from superficial gastritis with optimal diagnostic performance (area under the receiver operating curve [AUC] = 0.82) [69]. Finally, in a population of 268 early GC cases, the relative abundance of *H. pylori*, *Propionibacterium acnes*, and *Prevotella copri* in the stomach was higher than that of non-cancer-bearing people, thus identifying these taxa as strong risk factors for GC development. In contrast, *Lactobacillus lactis* was protective against GC [70].

Despite the importance of the findings, all previous studies are affected by inherent bias stemming from the comparison between different patient cohorts, specifically those with GC and cancer-free individuals. In such cases, variations in environmental factors and genetic backgrounds can significantly influence the composition of the microbiota. Therefore, some researchers chose a paired design approach, wherein the gastric microbiota was analyzed comparatively in the same patient’s tumor and non-malignant tissues.

Once again, the bacterial community differed considerably from non-tumoral and tumoral tissue in GC patients [71,72,73,74]. 

In cancer tissue, *H. pylori*, *Propionibacterium* spp., *Staphylococcus* spp., and *Corynebacterium* spp. were significantly reduced compared to normal gastric mucosa, while *Clostridium* spp. and *Prevotella* spp. were significantly increased [72]. Oral bacteria such as *Peptostreptococcus*, *Streptococcus*, and *Fusobacterium* were predominant among the enriched taxa in GC samples. At the same time, there were elevated levels of lactic acid-producing bacteria in the adjacent non-tumor tissues, including *Lactococcus lactis* and *Lactobacillus brevis* [73]. In a retrospective cohort study including 276 patients with GC, Liu et al. found that the bacterial richness was low in both the peritumoral and tumoral microhabitat. However, *H. pylori*, *Prevotella copri*, and *Bacteroides uniformis* significantly decreased, and *Prevotella melaninogenica*, *Streptococcus anginosus*, and *Propionibacterium acnes* significantly increased in the tumoral microhabitat. The specific stomach microenvironment, i.e., distinct pH, oxygen, nutrients, ions, and chemicals of tumor tissues and adjacent tumor-free tissues, were the main determinant of the composition and diversity of the gastric microbiota, providing a setting for symbiotic interactions among the various microbes within that ecosystem and the host [74]. 

Finally, the gastric microbiota profile differs significantly between early GC and advanced GC. Wang et al. demonstrated that *Burkholderia*, *Tsukamurella*, *Uruburuella*, and *Salinivibrio* varied in their relative abundance between the early and advanced stages of GC. Functional analyses showed that urease and bacterial flagella synthesis were impaired in early GC, whereas fructose glycolysis and glycoside hydrolysis were enhanced. Interestingly, the microbial signature could accurately distinguish early GC from chronic gastritis or advanced GC in an independent cohort of patients [75].

## 5. Microbiota Changes during the Multistep Process of Gastric Carcinogenesis

Changes in the gastric microenvironment, alterations in host immune responses, and the progression of GC can account for the different microecology of the stomach. In the early GC, the microbiota composition might reflect the initial stages of tumor development, with specific bacterial species potentially promoting or inhibiting tumor growth. As cancer progresses to advanced stages, the tumor microenvironment undergoes further changes, which can influence the composition of the gastric microbiota. Additionally, treatments such as chemotherapy and radiation therapy can impact microbiota composition. Overall, the dynamic interplay between the host, the tumor, and the microbiota likely contributes to the observed differences between early and advanced GC.

Notably, the gastric microbiota undergoes diversity and composition alterations also through the different stages of the gastric precancerous cascade [76], suggesting that dysbiosis worsens as the carcinogenesis process advances.

Aviles-Jimenez et al. first demonstrated a gradual shift in microbiota composition from NAG to intestinal metaplasia and GC. In particular, bacterial diversity steadily decreased as patients progressed from NAG to intestinal metaplasia and GC. At the genus level, *Porphyromonas*, *Neisseria*, TM7, and *Streptococcus sinensis* had a decreasing trend, while *Lactobacillus coleohominis* and *Lachnospiraceae* increased, suggesting that these alterations might favor gastric tumorigenesis. Interestingly, the microbiota segregated between NAG and GC, with 44 taxa showing significant changes in relative abundance [77]. 

In another study, GC patients had an increase in the *Bacilli* class and *Streptococcaceae* family and a reduced *Epsilonproteobacteria* class and *Helicobacteraceae* family compared to those with chronic gastritis and intestinal metaplasia [78]. 

To what extent changes in gastric microbiota are influenced by histological alterations of the mucosa or the presence of *H. pylori* is not entirely understood. 

Li et al. characterized the microbiota alterations through different histological stages of gastric carcinogenesis and after *H. pylori* eradication. Apart from dominant *H. pylori*, *Proteobacteria*, including *Haemophilus*, *Serratia*, *Neisseria*, and *Stenotrophomonas*, were the major components of the human gastric microbiota. Interestingly, there was a strong negative correlation between the relative abundance of *H. pylori* and bacterial diversity. However, in GC patients, there was a more pronounced decreasing trend in bacterial diversity compared to other precancerous lesions with comparable levels of *H. pylori*. Interestingly, eradicating *H. pylori* infection restored the gastric microenvironment to that of *H. pylori*-negative subjects, including a similar phyla composition and increased bacterial diversity index [79]. In *H. pylori*-negative cases, across precancerous stages from atrophic gastritis to dysplasia, *Burkholderiaceae* abundance constantly increased, while the abundance of *Streptococcaceae* and *Prevotellaceae* had a continuous decreasing trend. Patients with intestinal metaplasia and those with dysplasia had similar gastric mucosa microbiota profiles, with *Ralstonia* and *Rhodococcus* as the predominant genera. In contrast, GC had the lowest bacterial richness, with *Streptococcaceae* and *Lactobacillaceae* being the associated bacteria. Gastric juice had a higher microbial diversity than the mucosa, with *Proteobacteria* prevalent in the gastric mucosa and *Firmicutes* in gastric juice [27]. 

Comprehensive profiling of gastric mucosa and gastric juice showed a notable shift in microbial composition in both samples and a convergent dysbiosis during the progression from superficial gastritis to GC. This suggests that the colonization of carcinogenic microbes in the mucosa might originate from gastric juice and underscores the potential of gastric juice as a promising source of biomarkers [80,81]. Interestingly, a panel of GC classifiers, including *Gemella*, *Haemophilus*, *Peptostreptococcus*, *Streptococcus*, and *Veillonella*, exhibited considerable changes and performed well in an independent cohort, proving their effectiveness in identifying GC [80].

Relevant studies addressing the difference in gastric microbiota composition across the histological stages of gastric carcinogenesis are summarized in Table 1. 

However, differences in analytical methods, bioinformatic techniques, sampling strategies, participant demographics, target regions of the 16S rRNA gene, and sequencing platforms contribute to significant variability in the results, preventing the generalizability of the conclusions [85,86].

A meta-analysis explored the global microbial signature associated with gastric carcinogenesis by pooling data from six independent studies, including 825 gastric biopsy samples. Five phyla, *Proteobacteria*, *Firmicutes*, *Bacteroidetes*, *Actinobacteria*, and *Fusobacteria*, dominated the gastric microbiota in descending order of relative abundance. At the genus level, ten genera, namely *Helicobacter*, *Halomonas*, *Pseudomonas*, *Streptococcus*, *Lactobacillus*, *Shewanella*, *Prevotella*, *Acinetobacter*, *Cryocola*, and *Staphylococcus*, dominated the gastric mucosal microbiota. The alpha diversity decreased along the disease stages, with GC showing the lowest alpha diversity compared to superficial gastritis, atrophic gastritis, and intestinal metaplasia. Interestingly, the microbial composition of the four disease stages significantly clustered according to beta-diversity indexes. Opportunistic pathogens such as *Fusobacterium*, *Parvimonas*, *Veillonella*, *Prevotella*, and *Peptostreptococcus* were enriched in GC. At the same time, the commensals *Bifidobacterium*, *Bacillus*, and *Blautia* were depleted in GC compared to superficial gastritis. Overall, six GC-enriched (*Veillonella*, *Dialister*, *Granulicatella*, *Herbaspirillum*, *Comamonas*, and *Chryseobacterium*) and two GC-depleted genera (*Shewanella* and *Helicobacter*) were identified as potential biomarkers for discriminating GC from superficial gastritis, yielding an outstanding diagnostic performance (AUC = 0.91 for the training set and 0.85 for the test set, respectively). Except for *Veillonella*, *Herbaspirillum*, and *Shewanella*, all the evaluated biomarkers significantly differed between GC and superficial gastritis in Asian and European populations. Moreover, *H. pylori*-positive samples exhibited reduced microbial diversity, altered microbiota community, and weaker interactions among gastric microbes [87]. 

A more recent meta-analysis of nine public datasets of human gastric microbiota confirmed that the gastric microbiome composition significantly shifted during the progression of gastric carcinogenesis, accompanied by a gradual decline in microbial diversity. Using the random effect model, the authors identified a set of universal microbial biomarkers that differentiate GC and gastritis and have the potential for GC diagnosis. Key microbial signatures included *Fusobacterium*; *Leptotrichia*; and several lactic acid bacteria, such as *Bifidobacterium*, *Lactobacillus*, and *Streptococcus anginosus* [20].

In summary, the microbiota profiles in patients with *H. pylori*-induced superficial gastritis or glandular atrophy are characterized by the dominance of *Helicobacter* and, to a much lesser extent, *Streptococcus*, *Prevotella*, and *Neisseria* with a decreased phylotype richness, diversity, and evenness when compared with patients with normal gastric mucosa. The loss of specialized glandular tissue and reduced acid secretion in GC tissue results in *H. pylori* loss and the enrichment of intestinal commensals, including *Lactobacillus*, *Enterococci*, *Carnobacterium*, *Parvimonas*, *Citrobacter*, *Clostridium*, *Achromobacter*, and *Rhodococcus*, as well as the oral species *Fusobacterium nucleatum*, *Veillonella*, *Leptotrichia*, *Haemophilus*, and *Campylobacter* [63].

As expected, functional alterations accompany the progressive dysbiosis through gastric disease stages. The samples from GC patients were significantly enriched in many metabolic pathways, including those related to purine metabolism [73], carbohydrate metabolism (such as short-chain fatty acid production) [63,67] or glycolysis and glycoside hydrolysis [75], lipopolysaccharide (LPS) and L-arginine biosynthesis [27], amino acid and nitrate metabolism [17,63], and peptidoglycan biosynthesis [20]. These findings align with the increased energy metabolism and concentration of nitrogen-containing compounds in the tumor microenvironment [73]. In contrast, pathways involved in the fermentation of short-chain fatty acids and branched amino acid metabolism were more abundant in superficial gastritis [27].

Overall, the data that emerged from both animal and human studies do not definitively clarify whether the microbial changes observed in GC are causative in the development of the disease or if they result from the histologic progression through the precancerous cascade. Although a unique profile of gastric microbiota associated with GC development remains elusive, numerous taxa, including members of the oral and intestinal microbial communities, are reported to be enriched in the gastric microbial communities of cancer patients. This suggests a complex evolution of gastric carcinogenesis potentially coordinated by *H. pylori* in cooperation with other microbial species contributing to the process (Figure 1).

## 6. Effects of *H. pylori* Eradication on the Gastric Microbiome and Patient’s Outcome

The data on the effect of *H. pylori* eradication on gastric microbiome are somewhat conflicting. 

A recent meta-analysis including nine studies and 546 participants showed that *H. pylori* eradication was associated with increased alpha diversity and restored the composition of commensals commonly dominant in the healthy stomach, independently of the therapeutic schedule used and length of follow-up [88]. Since the reduction of commensal microbial communities is a hallmark of GC risk, it is conceivable to hypothesize that restoring the abundance of *Firmicutes*, *Bacteroides*, and *Actinobacteria* might be one of the mechanisms by which *H. pylori* eradication reduces GC risk. 

On the other hand, in patients with endoscopic follow-up for >1 year, eradicating *H. pylori* infection led to the predominance of proinflammatory *Acinetobacter* in gastric corpus mucosa with a decrease in microbial diversity in about half of the cases [89]. Moreover, after *H. pylori* eradication, the abundance of some bacteria predominantly of oral origin, such as *Actinomyces*, *Granulicatella* or *Peptostreptococcus*, was associated with precancerous gastric lesions, thus significantly increasing the risk of GC [90].

Based on the available evidence, a careful assessment of the individual benefit–risk ratio should replace the unconditional “test-and-treat” strategy in eradicating *H. pylori* infection. 

## 7. Conclusions and Perspectives

The stomach hosts a complex core microbiota, and *H. pylori* and non-*Helicobacter* gastric microbiota might be interactive contributors rather than acting autonomously in the malignant alteration of gastric tissues. Similar to the driver–passenger hypothesis proposed in colorectal cancer and gut microbiota interactions [91], *H. pylori* could trigger the gastric carcinogenic process. At the same time, other bacteria may damage gastric mucosa, thus promoting the progression of gastric precancerous lesions to GC. 

Although progress has been made in characterizing the human gastric microbiota in recent years, definitive conclusions are hindered by the inherent biases of the current literature. Evidence mainly derives from case-control studies comparing gastric microbiota signatures from GC patients and healthy individuals. Additionally, most studies are limited by small sample sizes, heterogeneous enrolled populations, and cancer diagnoses that do not differentiate between histological types or previous treatments. Finally, research on the gastric virome and mycome is less extensive than bacterial studies, and only in recent years has it gained interest. 

At the current state of our knowledge, the most critical question remains whether the alterations in the composition of the gastric microbiota are causal or consequential to the development of gastric precancerous lesions and GC so that we cannot infer a causal relationship between dysbiosis and the development and progression of GC. In addition, definitive conclusions cannot be drawn on the beneficial or harmful effects of *H. pylori* eradication on gastric microbiota. 

Through in-depth research, considering large-scale, multicenter, cohort prospective studies with long-term follow-up could significantly contribute to elucidating the dynamic changes in the microbiota along the gastric carcinogenic process and possibly identify specific microorganisms that are potentially useful as biomarkers for GC diagnosis and prognosis. Furthermore, a complex biological process such as GC should be studied holistically, using an integrative approach that combines multi-omics data to highlight the interrelationships of the involved biomolecules and their functions.

An integrative approach could clarify the mechanisms by which microbes impact cancer development and prognosis. Finally, a better understanding of the impact of eradicating *H. pylori* infection on gastric microecology could help to personalize indication to therapy. As our knowledge advances, new avenues for the early detection and treatment of GC will become available, offering guidance for determining personalized treatment strategies and improving patient outcomes. 

## Figures and Tables

**Figure 1 antibiotics-13-00484-f001:**
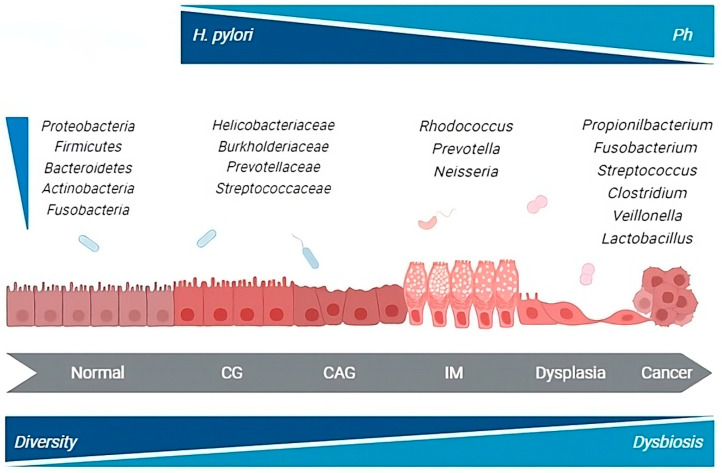
*H. pylori* and gastric microbiota alterations throughout the carcinogenic process. CG, chronic gastritis; CAG, chronic atrophic gastritis; IM, intestinal metaplasia. Gastric microbiota changes through the different steps of the carcinogenic cascade. GC-associated microbiota is enriched in species from the oral and intestinal microbial communities that may possess nitrosation functions.

**Table 1 antibiotics-13-00484-t001:** Changes in gastric microbiota in carcinogenesis.

Author	Population	Country of Origin	Sample Type	Sequencing Methods	Results
Dicksved et al. [66]	GC (10) vs. FD (5)	Sweden	Biopsy samples from the antrum and corpus	T-RFLP, 16S rRNA sequencing	↓ *H. Pylori* ↑ *Streptococcus*, *Lactobacillus*, *Veillonella* and *Prevotella*
Aviles-Jimenez et al. [77]	NAG (5), IM (5), GC (5)	Mexico	Biopsy samples from the antrum and corpus	Microarray G3 PhyloChip (16sRNA V3)	GC ↓ *Porphyromonas*, *Neisseria* and TM7 ↑ *Lachnospiraceae* and *Lactobacillus coleohominis*
Eun et al. [78]	GC (11), IM (10), CG (10)	Korea	Biopsy samples	16S rRNA sequencing V5	GC ↑ *Bacilli class* and *Streptococcaceae* family ↓ *Epsilonproteobacteria* class and *Helicobacteriaceae* family
Castaño-Rodríguez et al. [67]	GC (12), vs. FD (20)	Singapore and Malaysia	Biopsy samples from the antrum	16S rRNA sequencing V4	↑ *Lactococcus*, *Veillonella*, *Haemophilus*, and *Fusobacteriaceae*
Ferreira et al. [17]	GC (54) vs. CG (81)	Portugal	Biopsy samples	16S rRNA sequencing V5–V6	↓ *H. Pylori* ↑ *Achromobacter*, *Citrobacter*, *Clostridium*, *Lactobacillus*, and *Rhodococcus*
Coker et al. [69]	GC (20), IM (17), AG (23), SG (21) Validate: GC (19), AG (51), SG (56)	China	Biopsy samples from the antrum, body, and fundus	16S rRNA sequencing V4	GC ↑ *Parvimonas micra*, *Dialister pneumosintes*, *Slackia exigua*, *Peptostreptococcus stomatis*, *Prevotella intermedia*, *Fusobacterium* *nucleatum*, *Prevotella oris*, and *Catonella morbi*
Li et al. [79]	CG (9), IM (9), GC (7), *H. pylori* (−) control (8)	Hong Kong	Biopsy samples from the antrum and corpus	16S rRNA sequencing V3–V4	GC ↑ *Flavobacterium*, *Klebsiella*, *Serratia marcescens*, *Stenotrophomnonas*, *Achromobacter*, and *Pseudomonas*
Sun et al. [27]	*H. pylori* (−) CG (56), *H. pylori* (−) AG (9), *H. pylori* (−) IM (27), *H. pylori* (−) Dys (29), *H. pylori* (−) GC (13).	China	Biopsy samples and gastric juice	16S rRNA sequencing V3–V4	From AG to Dys ↑ *Burkholderiaceae* *↓ Streptococcaceae* and *Prevotellaceae* GC ↑ *Streptococcaceae* and *Lactobacillaceae*
He et al. [80]	GC (61), IM (55), GC (64)	China	Biopsy samples and gastric juice	16S rRNA sequencing V4	GC classifiers in both GM and GF, including *Lactobacillus*, *Veillonella*, *Gemella*
Gunathilake et al. [70]	GC (268) vs. Controls (288)	Korea	Biopsy samples	16S rRNA sequencing V3–V4	↑ *Propionibacterium acnes*, *Prevotella copri* ↓ *Lactococcus lactis*
Hu et al. [82]	SG (5), GC (6)	China	Gastric wash samples	Shotgun metagenomic sequencing	↓ Sphingobium yanoikuyae, ↑ *Aggregatibacter*, *Alloprevotella*, and *Neisseria*
Hsieh et al. [83]	CG (9), IM (7), GC (11)	Taiwan	Biopsy samples	16S rRNA sequencing V3–V4	↓ *H. Pylori* ↑ *Fusobacterium*, *Lactobacillus* and *Clostridium*
Yu et al. [71]	GC (160)	China and Mexico	Biopsy samples from non-malignant and tumor tissues	16S rRNA sequencing V3–V4	GC is dominated by *Proteobacteria*, *Bacteroidetes* (Chinese), or *Firmicutes* (Mexican) *H. pylori* abundance is lower in tumor tissue compared to matched non-malignant tissue
Liu et al. [74]	GC (276)	China	Biopsy samples from non-malignant and tumor tissues	16S rRNA sequencing V3–V4	↓ *H. pylori*, *Prevotella copri*, and *Bacteroides uniformis* ↑ *Prevotella melaninogenica*, *Streptococcus anginosus*, and *Propionibacterium acnes*
Park et al. [84]	CG (16), GAD (16), EGC (36), AGC (20)	Korea	Gastric juice samples	16S rRNA sequencing V3–V4	CG ↑ *Akkermansia* and *Lachnospiraceae NK4A136 Group* GC ↑ *Lactobacillus* and *Veillonella*.

GC, gastric cancer; FD, functional dyspepsia; NAG, non-atrophic gastritis; SG, superficial gastritis; AG, atrophic gastritis; IM, intestinal metaplasia; Dys, dysplasia; CG, chronic gastritis; GAD, gastric adenoma; EGC, early gastric cancer; AGC, advanced gastric cancer.

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
