# Peer review of "Exploring the Link between Helicobacter pylori, Gastric Microbiota and Gastric Cancer"

_antibiotics, 2024, doi:10.3390/antibiotics13060484_

Round 1

Reviewer 1 Report

Comments and Suggestions for Authors

A really interesting, well-organized manuscript concerned with a really important issue in gastric cancer: the role of microbiota richness (including H. pylori) during the carcinogenesis process. Only there are some minor issues that the authors can clarify easily:

1.       In Line 121, what SPF-GAN means? In 122, what PF-GAN means?

2.       The paragraph from lines 152 to 159 is out of context. Authors have to explain why they chose Colombia to give examples of differences in gastric microbiota.  

3.       Change “Columbia” in line 152 for “Colombia”.

4.       In line 171, even when the authors provide the reference, would be easier to understand the sentence if they specify that CG means chronic gastritis.

5.       The paragraph from lines 225 to 234 fits better in 4. Microbiota changes during the multistep process of gastric carcinogenesis.

6.       In the Table 1 footnote SG is missing.

Comments on the Quality of English Language

A really interesting, well-organized manuscript concerned with a really important issue in gastric cancer: the role of microbiota richness (including H. pylori) during the carcinogenesis process. Only there are some minor issues that the authors can clarify easily:

1.       In Line 121, what SPF-GAN means? In 122, what PF-GAN means?

2.       The paragraph from lines 152 to 159 is out of context. Authors have to explain why they chose Colombia to give examples of differences in gastric microbiota.  

3.       Change “Columbia” in line 152 for “Colombia”.

4.       In line 171, even when the authors provide the reference, would be easier to understand the sentence if they specify that CG means chronic gastritis.

5.       The paragraph from lines 225 to 234 fits better in 4. Microbiota changes during the multistep process of gastric carcinogenesis.

6.       In the Table 1 footnote SG is missing.

I consider that once the authors resolve these issues, the manuscript can be ready to be accepted in the Antibiotics Journal.

Reviewer 2 Report

Comments and Suggestions for Authors

This review by Costantino Sgamato et al. analyzed the current state of knowledge regarding the relationship between H. pylori and non-Helicobacter gastric microbiota in gastric carcinogenesis, highlighting the mechanisms by which microecological dysbiosis can contribute to the malignant transformation of the mucosa. The authors note that H. pylori could act as a trigger in the gastric carcinogenic process and other bacteria may damage gastric mucosa, thus promoting the progression of gastric precancerous lesions to GC. The authors conclude that large-scale, multicenter, prospective studies could significantly contribute to elucidating the dynamic changes of the microbiota along the gastric carcinogenic process and possibly identify specific microorganisms potentially useful as biomarkers for gastric cancer development and prognosis. The topic is valid, and the work is scientifically sound.  This is a comprehensive description on the relationship between H pylori and gastric carcinogenesis. The manuscript is well written overall and adequate references are included. The manuscript can be improved by addressing the following concerns.

-       Discuss in detail about the interplay between dietary factors, socioeconomic status, Westernization and other environmental factors that can affect the relationship between H pylori and gastric carcinogenesis.

-       Authors may also want to discuss about role of single nucleotide polymorphisms and increased risk of H pylori related GC.

-       Add limitations of this review in the Discussion section.

-       Include a section on gaps in our knowledge and recommendations for further research.

Reviewer 3 Report

Comments and Suggestions for Authors

In the manuscript “Exploring the link between Helicobacter pylori, gastric microbiota and gastric cancer”, the authors present and discuss the differences and roles of microbiota, with an emphasis on H.pylori, in gastric malignization. By taking into consideration the most recent evidences, the authors summarized the most common genera from healthy to malignant tissue, taking into consideration the different stages of pre-malignant lesions.

Suggestions:

Line 152: Colombia.

Line 171: define “CG” the first time it appears.

Line 236: avoid eponyms. Include citation (DOI: 10.1111/j.1751-2980.2011.00550.x)

Line 375: provide reference (DOI: 10.1038/nrmicro2819)

Comments on the Quality of English Language

Minor editing of English language required
